# Potential Wound Healing Effect of Gel Based on Chicha Gum, Chitosan, and *Mauritia flexuosa* Oil

**DOI:** 10.3390/biomedicines10040899

**Published:** 2022-04-14

**Authors:** Maria Onaira Gonçalves Ferreira, Alessandra Braga Ribeiro, Marcia S. Rizzo, Antonia Carla de Jesus Oliveira, Josy Anteveli Osajima, Leticia M. Estevinho, Edson C. Silva-Filho

**Affiliations:** 1Graduate Program in Materials Science, Campus Universitario Ministro Petrônio Portella, Federal University of Piaui, Teresina 64049-550, PI, Brazil; mariaonaira@gmail.com (M.O.G.F.); marciarizzo@ufpi.edu.br (M.S.R.); a.carlinha.18@gmail.com (A.C.d.J.O.); josyosajima@ufpi.edu.br (J.A.O.); 2CBQF–Centre of Biotechnology and Fine Chemistry–Associate Laboratory, Faculty of Biotechnology, Catholic University of Portugal, Rua Diogo Botelho 1327, 4169-005 Porto, Portugal; abribeiro@porto.ucp.pt; 3Mountain Research Center, CIMO, Polytechnic Institute of Bragança, Campus Santa Apolónia, 5300-253 Bragança, Portugal

**Keywords:** polysaccharides, chicha gum, chitosan, *Mauritia flexuosa* oil, antimicrobial activity, wound healing

## Abstract

Wounds are considered a clinically critical issue, and effective treatment will decrease complications, prevent chronic wound formation, and allow rapid healing. The development of products based on naturally occurring materials is an efficient approach to wound healing. Natural polysaccharides can mimic the extracellular matrix and promote cell growth, thus making them attractive for wound healing. In this context, the aim of this work was to produce a gel based on chicha gum, chitosan, and *Mauritia flexuosa* oil (CGCHO) for wound treatment. TG and DTG analyzed the thermal behavior of the materials, and SEM investigated the surface roughness. The percentages of total phenolic compounds, flavonoids, and antioxidants were determined, presenting a value of 81.811 ± 7.257 µmol gallic acid/g *Mauritia flexuosa* oil, 57.915 ± 0.305 µmol quercetin/g *Mauritia flexuosa* oil, and 0.379 mg/mL, respectively. The anti-inflammatory was determined, presenting a value of 10.35 ± 1.46% chicha gum, 16.86 ± 1.00% *Mauritia flexuosa* oil, 10.17 ± 1.05% CGCHO, and 15.53 ± 0.65% chitosan, respectively. The materials were tested against Gram-negative (*Klebsiella pneumoniae*) and Gram-positive (*Staphylococcus aureus*) bacteria and a fungus (*Candida albicans*). The CGCHO formulation showed better antimicrobial activity against Gram-positive bacteria. In addition, an in vivo wound healing study was also performed. After 21 days of treatment, the epidermal re-epithelialization process was observed. CGCHO showed good thermal stability and roughness that can help in cell growth and promote the tissue healing process. In addition to the good results observed for the antimicrobial, antioxidant, anti-inflammatory activities and providing wound healing, they provided the necessary support for the healing process, thus representing a new approach to the wound healing process.

## 1. Introduction

Wound healing is a dynamic and complex physiological process that occurs following a natural response to injury, in which some irregularities may occur. Research has been developed to accelerate the process of wound healing [1]. The healing process involves four phases: coagulation, inflammation, migration-proliferation, and remodeling. These phases occur in sequence; however, different stages can coincide in other areas of the injury site during the repair process. [2,3,4,5]. The hemostasis stage occurs immediately after the injury. The inflammation phase, characterized by signs of heat, redness, swelling, and pain, is fully established within 24 h, but it may last longer if there is an infection, trauma, or other problem. With their degradative enzymes and reactive oxygen species (ROS), phagocytes decompose and remove cell fragments at tissue damage sites [6]. The extracellular matrix (ECM) provides the structure for repair and is one of the main components of wound healing, along with stromal cells (fibroblasts, myofibroblasts). The healing stage can be promoted by pharmaceutical formulations containing antioxidant, antimicrobial, hemostatic, or anti-inflammatory compounds [7,8].

Many studies have been carried out to develop an ideal material to facilitate healing and provide the physiological environment for the wound [5]. Natural polymers can act as an accelerating agent in the wound healing process, as an anti-inflammatory agent, and as safe and biocompatible scaffolds for skin tissue regeneration. Polymers such as chitosan, gelatin, alginate, and hyaluronic acid have been used in wound healing [9,10]. These materials can adhere to the injured tissue through their surface molecules by means of chemical bonds, electrostatic bonds, and diffusive phenomena. The pattern of chemical bonds involved in these materials can also have an influence, as diffusion is believed to occur through the movement and penetration of polymer chains between the macromolecules of the cell membrane [11]. Materials such as polysaccharides have aroused great interest in the pharmacological sector. Some interesting properties can be found in some polysaccharides, especially biocompatible, biodegradable, adhesive, non-toxic, antimicrobial, antioxidant, and anti-inflammatory activities. Additionally, they have been used as materials with potential uses in the wound healing process [5,12,13,14,15,16].

Chitosan (CH), a polysaccharide obtained from crustacean shells, has a bioactive potential that partially meets the requirements for fast curing [17,18]. The structure of CH is similar to the ECM of the skin, composed of hyaluronic acid (HA), a polysaccharide that has N-acetyl-D-glucosamine groups in its structure [19,20]. CH hydrogels improved mucoadhesion properties and accelerated the healing rate by successfully reconstructing an epidermis [21,22]. CH promotes the growth of wound healing-related cells, such as the growth of fibroblasts, epithelial cells, keratinocytes, and macrophages to produce active factors that contribute to wound healing. In addition, the molecular structure of chitosan is similar to the stroma of mucopolysaccharide matrix cells [23,24,25]. Another biopolymer with interesting biological properties for treating skin lesions is chicha gum (CG), a polysaccharide from the exudate of the *Sterculia striata* tree. CG is an anionic polysaccharide, with numerous acid groups in its structure, with excellent swelling capacity, adhesion, anti-inflammatory, and antibacterial activity, important features for the wound healing process [26,27,28]. Due to its properties, GC has the potential to be explored for asset delivery systems. GC hydrogels have been developed and have shown firmness, consistency, and cohesiveness. The mucoadhesion of the CG hydrogel was superior to that of chitosan, being a promising polymer for mucoadhesive formulations [29,30].

*Mauritia flexuosa* oil is a natural activity extracted from the fruit of the *Mauritia flexuosa* L. palm, a common plant in the Amazon region of Brazil. This product is available in large- and small-scale production, acquisition, and processing [31]. Due to compounds such as fatty acids and carotenoids, this natural oil can help to heal wounds and acts as a healing accelerator [32]. Carotenoids, especially β-carotene, are antioxidant agents [24]. BO has fatty acids in its composition, such as oleic acid, which acts in the construction of the cell membrane and is one of the constituents of the epidermis [33,34,35,36,37]. Hexadecanoic and octadecanoic acids are other constituents of *Mauritia flexuosa* oil and tend to increase antimicrobial and antioxidant activity [38,39]. The antimicrobial activity occurs because fatty acids act as detergents in the amphipathic structure of a bacterial cell membrane, increasing cell permeability and impairing essential processes [40]. *Mauritia flexuosa* oil has the potential for use in the pharmaceutical and cosmetic industries. Oil emulsions were considered potential vehicles to transport antioxidant precursors. Thus, new products with *Mauritia flexuosa* oil can prevent pathologies associated with oxidative damage [41,42]

The properties offered by CH, CG and *Mauritia flexuosa* oil are attractive for wound treatment. Thus, this work aimed to synthesize a gel formulation of CH associated with CG and *Mauritia flexuosa* oil (CGCHO) for wound healing.

## 2. Materials and Methods

### 2.1. Materials

The reagents sodium hydroxide, sodium chloride, ethanol, methanol, acetone, acetic acid, hyaluronidase enzyme (350 UI), potassium salt of the human umbilical cord, hyaluronic acid, Trolox, quercetin, and 2,2-diphenyl-1-picryl-hydrazine were obtained from Sigma-Aldrich (St. Louis, MO, USA). Chitosan with a medium degree of deacetylation of 95% was supplied by Primex. *Mauritia flexuosa* oil was purchased at local markets in the city of Bom Jesus, State of Piaui, Brazil. Brain heart infusion and Mueller–Hinton agar were obtained fro, HIMEDIA (Mumbai, Maharashtra India). Sodium sulfate was provided by FLUKA (Steinheim, Renânia do Norte-Vestefália, Germany), and gallic acid was obtained from Panreac. Folin, sodium carbonate, and aluminum chloride were obtained from MERCK (Hohenbrunn, Baviera, Germany).

### 2.2. Purification of Chicha Gum

Crude CG samples were collected from bark-free nodules of chicha tree (*Sterculia striata*) planted at EMBRAPA-Meio Norte, located in Teresina, Piauí, Brazil. The material was deposited and registered by the Graziela Barroso Herbarium and received the following registration number, TEPB:30418. CG (Mw = 6.3 × 10^6^ g/mol) was isolated and purified according to the method proposed by Braz et al. [26]. The exudate (1.0 g) was dissolved in distilled water with stirring (100.0 mL) at room temperature for 24 h. Subsequently, 1 g of NaCl was added to the solution, and the pH was adjusted to 7. This polysaccharide was precipitated in ethanol P.A (99.5%). Later, the precipitate was washed with ethanol, followed by acetone, dried at 40 °C for 24 h, and ground until a powder was formed.

### 2.3. Preparation of Chicha Gum/Chitosan/Mauritia flexuosa Oil Gel (CGCHO) and Chitosan-Based Gel (CHB)

The CGCHO gel was fabricated using 1.0 g of CG, 1.5 g of CH, and 0.5 mL of *Mauritia flexuosa* oil in 25.0 mL of acetic acid. The gel was homogenized using a mechanical stirrer for 10 min. The oil-free gel was also obtained with the same concentrations of polysaccharides. The CHG gel was prepared by adding 0.75 g in 25.0 mL of acetic acid (2% *v*/*v*); the solution was stirred to form a homogeneous gel.

### 2.4. Characterization of Polysaccharides and Gels

#### 2.4.1. Fourier Transform Infrared Spectroscopy (FTIR) Analysis

FTIR spectra of the polysaccharides and gels were obtained using a Varian 660-IR spectrophotometer. The range was 600 to 4000 cm^−1^, combining 16 scans, with a resolution of 4 cm^−1^.

#### 2.4.2. Thermal Gravimetric Analysis (TGA)

TGA analysis was performed using TA Instruments SDT Q600 V20.9 Build 20, model DSC-TGA Standard (New Castle, DE, USA). Approximately 10.0 mg of each dried sample was placed in an aluminum crucible and heated from 25 to 600 °C at 10 °C min^−1^ in a nitrogen atmosphere.

#### 2.4.3. Scanning Electron Microscopy (SEM)

A freeze-dried gel sample was put on double-sided carbon tape and sputter-coated with silver. SEM images of samples were observed using an FEI, model Quanta FEG 250 (FEI Company, Eindhoven, Holanda). For the lyophilization process, the sample was frozen in an ultrafreezer (−80 °C), and it was not necessary to use cryoprotectant.

### 2.5. Characterization of Mauritia flexuosa Oil

#### 2.5.1. Antioxidant Activity of *Mauritia flexuosa* Oil-DPPH Radical Scavenging Assay

*Mauritia flexuosa* oil’s antioxidant activity was determined by the DPPH• [43]. The analysis used 20.0 μL of oil mixed with 180.0 μL of DPPH• solution (150 μmol/L) in methanol. The concentrations of the oil were 30.0 to 150.0 μL/mL. The mixture was incubated at room temperature for 40 min in the dark. The absorbance was measured at 517 nm. Trolox was used as a control. The experiments were performed in triplicate. The inhibition percent was calculated from the control using Equation (1):(1)$$\%\;\mathrm{DPPH}•=\left[1-\left(\frac{\mathrm{Abs}\;\mathrm{sample}}{\mathrm{Abs}\;\mathrm{control}}\right)\right]\times 100$$
where:

% DPPH• = activity of DPPH•

Abs sample = Absorbance at the sample

Abs control = Absorbance at the control

The concentrations of the *Mauritia flexuosa* oil were determined when 50% inhibition (IC_50_) was observed.

#### 2.5.2. Total Phenolics and Flavonoids of *Mauritia flexuosa* Oil

The total phenolic content (TPC) of *the Mauritia flexuosa* oil was determined using the Folin-Ciocalteu method [44,45]. Briefly, 20.0 μL of a dilute solution of oil in ethanol (10.0 μL/mL and 2.5 μL/mL) was mixed with 100.0 μL of Folin-Ciocalteu reagent. The solution was stirred for 1 min and kept at rest for 4 min. After, 75.0 μL of Na_2_CO_3_ (10% *w*/*v*) was added and stirred for 1 min. After incubation in the dark for 2 h, at room temperature, the absorbance of the solution and the blank were determined by UV-vis at 750 nm (VWR UV-3100 PC Spectrophotometer). The total phenolic content was calculated from a calibration curve, using gallic acid as standard. The result was expressed as milligrams of gallic acid equivalents per gram of *Mauritia flexuosa* oil.

To determine the total content of flavonoids in *Mauritia flexuosa* oil, the aluminum chloride method was used [45,46]. Briefly, 20.0 μL of a diluted solution of oil in ethanol (10.0 μL/mL and 2.5 μL/mL) was mixed with 210.0 μL of methanol (80%) and 20.0 μL of 2% AlCl_3_ was added. The solution was mixed and kept in the dark at room temperature for 30 min, and the absorbance of the mixture was monitored by UV-Vis at 427 nm (VWR UV-3100 PC Spectrophotometer). The total flavonoid content was expressed as milligrams of quercetin equivalents per gram of *Mauritia flexuosa* oil.

### 2.6. Anti-Inflammatory Activity of Polysaccharides and Gels

The anti-inflammatory activity was indirectly evaluated from the inhibitory effect of samples on reactions catalyzed by hyaluronidase [32,47]. The analysis was performed for samples of CG, *Mauritia flexuosa* oil, CGCHO and, CHG. Briefly, 50.0 μL of each sample was used in 50.0 μL (350 UI) of hyaluronidase enzyme (Type IV-S) incubated at 37 °C for 20 min. Then, calcium chloride (1.2 mL of 2.5 × 10^−3^ mol/L) was added to activate the enzyme, and the mixture was incubated for a further 20 min at 37 °C. After this time, 0.5 mL of hyaluronic acid sodium salt (1.0 × 10^−1^ mol/L) was added to each sample and incubated for another 40 min at 37 °C. Then, 0.1 mL of potassium tetraborate was added at a concentration of 8.0 × 10^−1^ mol/L. Then, the solutions were boiled in a water bath for 3 min. The solutions were cooled to 10 °C with the addition of 3.0 mL of p-dimethylaminabenzaldehyde and incubated for 20 min at 37 °C. Finally, absorbance was measured by UV-Vis at 585 nm. All tests were performed in triplicate.

### 2.7. Antimicrobial Tests

#### Minimum Inhibitory Concentration (MIC)

The experiments were conducted against three different microorganisms: *Staphylococcus aureus* (ATCC 43300), *Klebsiella pneumoniae* (ATCC 13883), and the fungus *Candida albicans* (ATCC 10231).

The minimum inhibitory concentration (MIC) was determined in a 96-well microplate, and the experiments were performed according to the CLSI [48]. In brief, isolated colonies were suspended in sterile saline with 0.85% to reach 0.5 on the MacFarland scale. The broths used were Muller–Hinton broth for bacteria and yeasts peptone dextrose broth for the fungus. CG, CGCHO, and CHG were tested at a concentration of 0.156 to 40.0 mg/mL. Isolated *Mauritia flexuosa* oil was also tested at a concentration of 2.5 mg/mL to 320.0 mg/mL. To all wells in the plates were added the isolated colonies previously prepared (10^6^ colony forming units (CFU)/mL) and incubated at 37 °C for 24 h and at 25 °C for 48 h for the bacteria and fungus, respectively. Then, the antimicrobial activity was determined by the addition of 20.0 μL of 2, 3, 5-triphenyl-2H-tetrazoliumchloride (TTC, 5 mg/mL) solution. The antibacterial and antifungal controls used were gentamicin and amphotericin, respectively. All tests were performed in triplicate.

### 2.8. Healing Assays

#### 2.8.1. Excision Procedure, Wound Treatment, and Skin Lesion Assessment in Rats

The wound excision procedure was described by Ferreira et al. (2020) [32]. The animals were anesthetized with xylazine hydrochloride (0.04 mL/100 g) and ketamine hydrochloride (10% 0.08 mL/100 g). Then, skin tissue approximately 0.6 cm in diameter was removed to form a wound with exposure of the dorsal fascia muscle. The animals were treated with topical application of 10.0 mg of each sample of CHB, *Mauritia flexuosa* oil and CGCHO.

Skin lesion evaluation in animals was performed, observing the repair of the lesion and the presence or absence of edema, exudate, and crust. Lesions were measured on the 1st, 3rd, 7th, 14th and 21st days of treatment. After the test, the animals were euthanized with sodium pentobarbital (10–15 mg/100 g). A skin fragment from the back of three animals from each group was dissected on the 3rd, 7th, 14th and 21st days of treatment. The injured skin samples were fixed in 10% buffered formalin for 24 h and taken to a battery with an increasing sequence of alcohol and xylene. Then, they were subjected to paraffin baths at 60 °C and embedded in wax blocks. After this, 5 µm-thick sections were made using a Leica microtome (Buffalo Grove, IL, USA), stained with hematoxylin-eosin (H.E.). The slides were evaluated by light microscopy and the images were digitally photographed.

#### 2.8.2. Statistical Analysis

Data were analyzed using Graph Pad Prism Version 6.0 software, and the results of healing on the 3rd, 7th, 14th and 21st days of treatment were expressed by the significance of the difference between the mean using variance analysis (ANOVA) with *p* ≤ 0.05.

## 3. Results and Discussion

### 3.1. Characterizations

Figure 1A–D show the FTIR spectra of CG, *Mauritia flexuosa* oil, CGCHO and CHG, respectively. Figure 1A shows bands in the region of 3500 cm^−1^ related to the elongation of the galactopyranose and glucopyranose ring alcohols, and the band at 1643 cm^−1^ indicates the presence of carboxylic acid groups (black arrow) [26,28]. In Figure 1B, the region above 3000 cm^−1^ is related to the hydroxyl groups of the acid groups present in the BO structure. The bands in the region of 2900 cm^−1^ are related to the CH bonds of the carbons (blue arrow) [32]. In the regions of 1700–1500 cm^−1^, the presence of carbonyls was observed, referring to the fatty acids in the BO structure [49]. In Figure 1C,D, the band observed at 3100 cm^−1^ is related to OH stretching with large broadening caused by gel formation [32,50]. Close to 1650 cm^−1^, bands of C=O were observed (black arrow). Thus, in Figure 1D, the combination of the main characteristic bands of the polymers and BO is observed.

In Figure 2, SEM images of the CGCHO are shown at different magnifications. An irregular and porous surface with surface roughness was observed (black arrow). The presence of roughness can facilitate cell growth and favor the tissue healing process, due to its similarity to the extracellular matrix of the skin [51]. The porosity of CGCHO can be favorable to the transfer of nutrients and oxygen to the wound cells. This process can contribute to vascularization, better adhesion of the gel to the skin and cell proliferation [32,52]. In Figure 2c,d, it can be seen that *Mauritia flexuosa* oil is dispersed throughout the gel, suggesting uniform incorporation.

In Figure 3, the TG and DTG curves are shown. For the CG (Figure 3A,E), two mass loss events were observed. The first event is related to water loss and the second event is related to the degradation of the main polysaccharide polymer chain, according to Braz et al. [26]. *Mauritia flexuosa* oil presented two events in the temperature range between 250 and 350 °C. The first event is related to the condensation of surface groups. The second refers to the total thermal decomposition (Figure 3B,F). For CGCHO, two decomposition events were observed. The first event that occurred at 100 °C is related to the loss of water from the polymers. The second event occurred between 400 and 600 °C and is related to *Mauritia flexuosa* oil (Figure 3C,G). CGCHO showed greater thermal stability, indicating a good interaction between biopolymers and *Mauritia flexuosa* oil; thus, the use of this material allows it to be used in wound healing without suffering significant mass loss up to 50 °C. CHB showed three decomposition events. The first event occurred in the range of 50 °C and was related to the loss of water and other volatile components. The second event occurred between 100 and 200 °C and was related to the condensation of the hydroxyl and amine groups. The last event indicated decomposition of the polysaccharide structure (Figure 3D,H) [49].

### 3.2. Antioxidant Activity

The IC_50_ value was determined for *Mauritia flexuosa* oil and is related to the amount of compound required to lower the initial DPPH• radical concentration by 50%. The *Mauritia flexuosa* oil had a value of 0.379 mg/mL, indicating good activity. One of the factors responsible for the antioxidant activity in *Mauritia flexuosa* oil is the presence of phenolic compounds and flavonoids, in addition to other components such as carotenoids. High oxidative stability can neutralize free radicals naturally produced by neutrophils at the site of inflammation and contribute to rapid healing [32,44,45]. One of the main factors that aggravate pathological conditions is oxidative stress, being responsible for damage and molecular alterations [32,43].

### 3.3. Total Phenolics and Flavonoids

The content of phenolic compounds found for *Mauritia flexuosa* oil was 81.811 ± 7.257 µmol gallic acid/g *Mauritia flexuosa* oil. Phenolic compounds act as hydrogen donors, being effective antioxidants and powerful free radical scavengers capable of reducing oxidative damage [53,54,55,56,57]. It is noteworthy that the biological properties of these natural antioxidants present in *Mauritia flexuosa* oil can contribute to rapid healing.

The content of flavonoid found was 57.915 ± 0.305 µmol quercetin/g *Mauritia flexuosa* oil. Flavonoids act in the inflammatory process by mediating macrophages and the production of pro-inflammatory molecules. These results demonstrate that *Mauritia flexuosa* oil can be an enhancer in the healing process.

### 3.4. Anti-Inflammatory Activity

For anti-inflammatory activity, the materials showed enzyme inhibition percentages of 10.35 ± 1.46% for CG, 16.86 ± 1.00% for *Mauritia flexuosa* oil, 10.17 ± 1.05% for CGCHO and 15.53 ± 0.65% CH, as shown in Figure 4. It was observed that the CG was responsible for the decreased anti-inflammatory activity of the CGCHO gel; this result corroborates Ferreira et al. [32]. The inflammatory process is complex, induced by different pathways, and occurs mainly in the first three days after tissue injury. Cell proliferation, mainly of macrophages and neutrophils, at the wound site has a pro-inflammatory and anti-inflammatory action [47]. The materials used to produce the gel, such as *Mauritia flexuosa* oil, have antioxidant activity that inhibits nitric oxide and reduces the inflammatory process [47,58]. CH is responsible for stimulating cell proliferation, which justifies its anti-inflammatory action [59,60]. Chang et al. [61] reported that CH inhibited the production of pro-inflammatory cytokines (TNF-α, IL-6). CG has biological properties similar to CH, and so a similar anti-inflammatory activity is expected [26]. Thus, the use of natural polymers to produce CGCHO gel has good anti-inflammatory activities and has the potential to help in the wound healing process.

### 3.5. Antibacterial Tests

Table 1 presents the results of the antimicrobial tests. CGCHO gel showed better MIC results for *S. aureus* when compared to CHG. *Mauritia flexuosa* oil and CG did not show antibacterial activity at the concentrations tested. The activity observed for the CGCHO gel may be related to the presence of CH [28,32]. The cationic nature of CH interacts with the negatively charged microbial cell membrane, which leads to extravasation of intracellular components, causing bacterial cell death [62,63]. *S. aureus* is a pathogenic bacterium widely known to cause infections based on biofilm formation in its hosts. Thus, the use of a gel that has antibacterial activity can promote the healing process.

The CGCHO gel showed a superior inhibitory effect than that observed for CHG for *Klebsiella pneumoniae*. *K. pneumoniae* is associated with a variety of infections, including wounds and disseminated infections. Babassu oil fibers with PLA were produced to aid in the healing process, and in the antimicrobial assay, only inhibition was observed for Gram-negative bacteria [64]. *Mauritia*
*flexuosa* oil and CG showed no antibacterial action against this strain at the concentrations tested. Between the two strains tested, it was possible to observe a better result for *S. aureus*. This result may be related to the structural differences in the cell wall of both bacteria. Gram-positive bacteria have only one cell membrane, whereas Gram-negative bacteria have a cell membrane and an additional outer layer membrane that is impermeable to most molecules.

The antifungal activity tested against *C. albicans* was observed for CGCHO and CHG gel. *C. albicans* is an opportunistic human pathogen involved in various infections and causes disease in immunocompetent and immunocompromised patients, accounting for a high mortality rate in the United States [65,66,67]. Antimicrobial action may help to reduce wound contamination by bacterial and fungal strains. Thus, CGCHO gel can be a viable alternative for the wound healing process.

### 3.6. Healing Activity

The in vivo assay for the treatment of skin wounds in mice was performed using the CGCHO. In Figure 5, the histopathological evaluations performed on days 3, 7, 14 and 21 of treatment are shown. Signs of the inflammatory phase were identified on the third day of treatment by histopathological analysis, such as the presence of ulcers, necrosis, and inflammatory response [68]. Inflammatory infiltrate was identified, with the presence of cellular debris and mononuclear cell infiltrate. In the deep dermis, mononuclear infiltrate, hemorrhage, fibrin network and cellular debris were observed in the wound (Figure 5A). On the seventh day of the experiment, the presence of fibrin, mononuclear cells, the formation of granulation tissue in the dermis and fibroblasts producing fundamental amorphous substances was observed. An angiogenesis process and some mast cells were observed. In the deeper dermis, young collagen fibers without individualization were observed. The re-epithelialization process was completed after 7 days of treatment with CGCHO, with a thicker epidermis close to the lesion area (Figure 5B). CH gel with nerolidol was produced and in the in vivo healing assays on day 7, the beginning of reepithelialization was also observed [3].

On the 14th day, the formation of capillaries and granulation tissue with dense collagen fibers and the presence of congested mononuclear cells were observed (Figure 5C). On the last day of treatment (day 21), the complete process of epidermal re-epithelialization was observed, with hair bulbs in the dermis, contractile fibroblasts, and denser and thicker collagen fibers (Figure 5D).

The healing process is a complex physiological reaction, being a dynamic procedure that requires the participation of different tissues and cell lines [69]. Events such as re-epithelialization and tissue repair play important roles in wound healing [70]. The healing properties observed in this study may be related to the characteristics of the CGCHO gel. CH has a structure similar to the glycosaminoglycans present in the extracellular matrix (ECM), having the ability to stimulate the re-epithelialization process [5,17,71]. CG is an anionic polysaccharide with swelling power and can facilitate wound repair [36,68]. *Mauritia flexuosa* oil is rich in fatty acids, has an immunomodulatory character, acting as a chemotactic agent and an angiogenesis promoter, and may interfere with the inflammatory process [23,24,25,26,27,28,29,30,31,32]. It has already been observed that extra virgin olive oil, *Moringa oleifera* oil, and *Melaleuca alternifolia* oil have demonstrated promising efficacy in the treatment, healing, and closure of wounds through their anti-inflammatory, antioxidant, antimicrobial properties and by stimulating the angiogenic activities necessary for wound healing [72,73,74]. The combination of biomaterials has a common interest because they act on molecular and cellular events that aim to repair the defect in tissue integrity [75,76]. Thus, the gel produced can provide moisture for re-epithelialization, and contains antibacterial agents, phenolic compounds and flavonoids that are antioxidant agents and act to control the production of pro-inflammatory molecules. In addition, it has adequate roughness to allow gas exchange, which can be useful in the healing process. However, since these are natural products, there is a need for further toxicological tests on the components of the CGCHO gel to demonstrate the efficacy and tolerance for topical treatment, as well as to identify synergy or antagonism between the components.

## 4. Conclusions

It was possible to develop a gel with chicha gum, chitosan and *Mauritia flexuosa* oil for the treatment of wounds. The gel showed interesting properties, such as roughness, antimicrobial activity against *Staphylococcus aureus* and *Klebsiella pneumoniae* and good anti-inflammatory activity. All of these properties probably favored the re-epithelialization process, maintaining a moist environment and stimulating the angiogenic activities necessary for wound healing, as observed in the in vivo study. However, further studies of toxicity, biodegradation profile and release profile of *Mauritia flexuosa* oil are needed for CGCHO gel. However, CGCHO gel may be a promising candidate for the treatment of wounds.

## Figures and Tables

**Figure 1 biomedicines-10-00899-f001:**
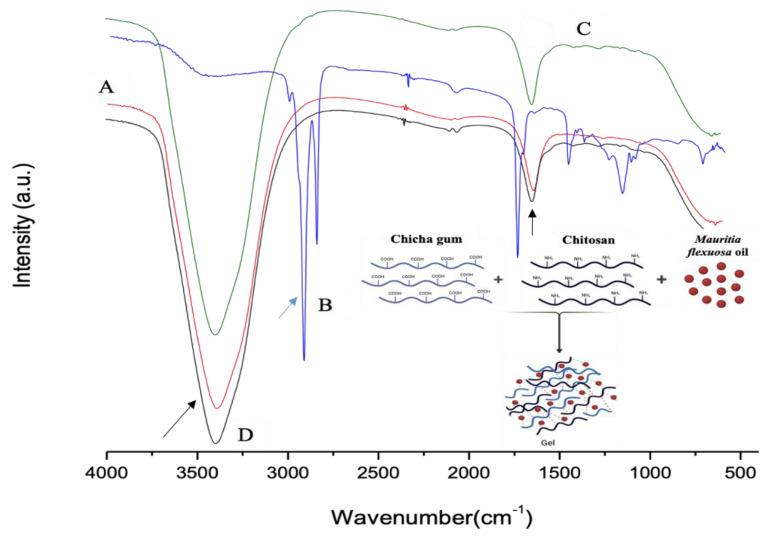
FTIR spectra for (A) chicha gum (CG), (B) *Mauritia flexuosa* oil, (C) chicha gum/chitosan/*Mauritia flexuosa* oil gel (CGCHO), and (D) chitosan-based gel (CHB).

**Figure 2 biomedicines-10-00899-f002:**
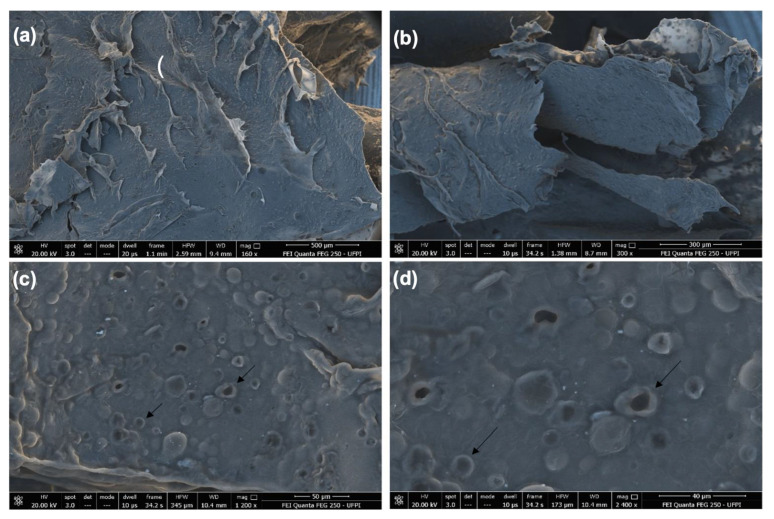
Scanning electron microscope (SEM) images of chicha gum/chitosan/*Mauritia flexuosa* oil gel (CGCHO) at the magnitudes of (**a**) 500 µm, (**b**) 300 µm, (**c**) 50 µm and (**d**) 40 µm.

**Figure 3 biomedicines-10-00899-f003:**
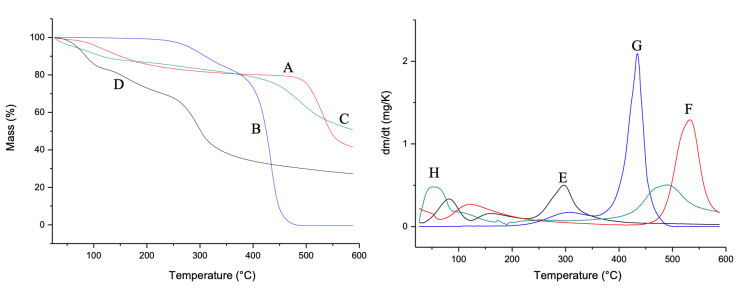
Thermogravimetric curves (TG) of (A) chicha gum (CG), (B) *Mauritia flexuosa* oil, (C) chicha gum/chitosan/*Mauritia flexuosa* oil gel (CGCHO), and (D) chitosan-based gel (CHB). Derivative thermogravimetric curves (DTG) of CG (E), *Mauritia flexuosa* oil (F), CGCHO (G), and CHG (H).

**Figure 4 biomedicines-10-00899-f004:**
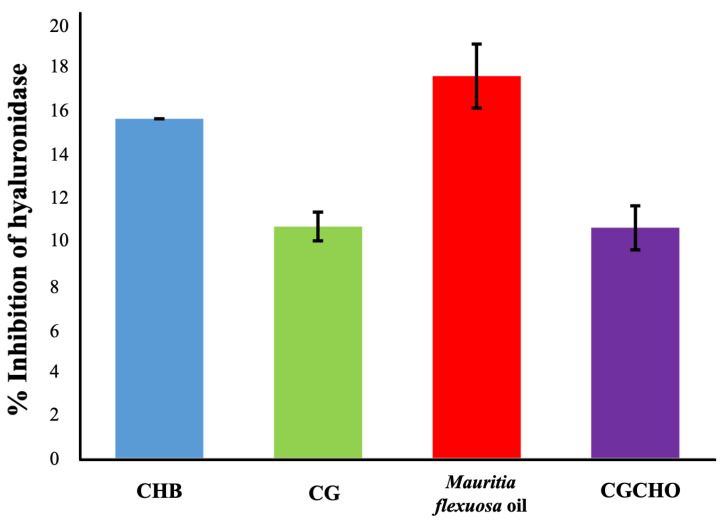
Inhibition of hyaluronidase activity by chicha gum (CG), *Mauritia flexuosa* oil, chicha gum/chitosan/*Mauritia flexuosa* oil gel (CGCHO), and chitosan-based gel (CHB).

**Figure 5 biomedicines-10-00899-f005:**
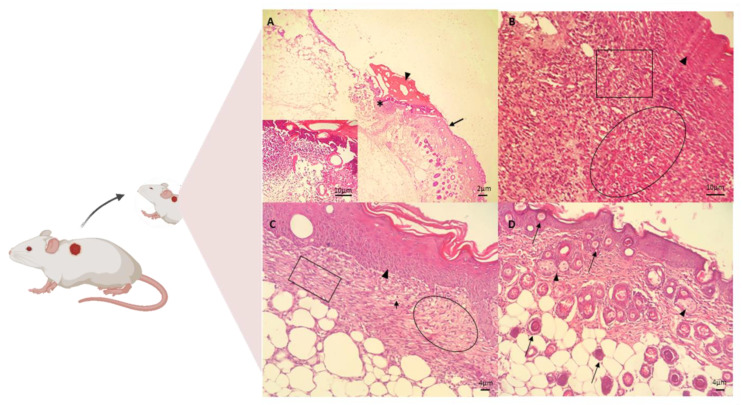
Photomicrographs of skin wounds on the (**A**) 3rd, (**B**) 7th, (**C**) 14th, and (**D**) 21st day of treatment for the group treated with chicha gum/chitosan/*Mauritia flexuosa* oil gel (CGCHO). H.E. stain. (Scale bar: 2 µm, 4 µm, and 10 µm). (**A**) Skin ulcer (*) with the epidermis on the right edge (arrows), fibrin clot (arrowheads), and inflammatory infiltrate of polymorphonuclear cells in the dermis (detail). (**B**) Continuing reepithelialization of the epidermis (arrowhead), presence of newly formed vessels in granulation tissue in the dermis (circle), and macrophages and new fibroblasts (square). (**C**) Epidermal reepithelialization (arrowheads) and extensive granulation tissue formation (circle), with neoformed vessels (arrows), and collagen fibers(square). (**D**) Presence of hairs follicle (bulbus-arrows) and sebaceous glands (arrowheads).

**Table 1 biomedicines-10-00899-t001:** Minimum inhibitory concentration (mic) for CG, *Mauritia flexuosa* oil, CGCHO and CHB against *S. aureus*, *K. pneumoniae* and *C. albicans.*

	*Staphylococcus aureus* ATCC 43300	*Klebsiella pneumoniae* ATCC 13883	*Candida albicans* ATCC 10231
**CG (mg/mL)**	-	-	-
***Mauritia flexuosa* oil (mg/mL)**	-	-	-
**CGCHO (mg/mL)**	2.50 ± 0.00	5.00 ± 0.00	10.00 ± 0.00
**CHB (mg/mL)**	10.00 ± 0.00	10.00 ± 0.00	10.00 ± 0.00
**Gentamicin (μg/mL)**	2.34 ± 0.54	5.87 ± 0.06	-
**Amphotericin B (μg/mL)**	-	-	0.96 ± 0.10

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
