# Peer review of "Potential Wound Healing Effect of Gel Based on Chicha Gum, Chitosan, and *Mauritia flexuosa* Oil"

_biomedicines, 2022, doi:10.3390/biomedicines10040899_

Round 1
Reviewer 1 Report
Manuscript ID: biomedicines-1660756 This manuscript, Potential wound healing effect of gel based on chicha gum, chitosan, and Mauritia flexuosa oil, is an interesting paper focusing on a novel gel based on two biopolymers incorporated with a bioactive oil. The paper is relatively well-thought, suitable for readers and shows new research results regarding wound healing. Although there are several points that should be improved:[1] The abstract should contain more precisely described the main effects of the research. [2] Keywords should be also more precise. [3] In the Introduction, there is no presentation of other biomaterials dedicated to the treatment of wounds; the use of this oil in biomaterials; summaries of research to date on CH and CG gels. [4] You stated: ,,[Polysacharides] They are biocompatibile, biodegradable, adhesive, non-toxic, antimicrobial, antioxidant and anti-inflammatory" - all of them? [5] There is no description of the statistical analysis method. [6] In the Materials and Methods, description of freeze-drying procedure is missing; justification for the selection of these bacteria / fungi supported by the literature. [7] In the Fig. 1, please also mark the describing in text groups (bonds, stretch etc.) [8] From these SEM images (Fig. 3), we are not able to determine adequately accurately the porosity of the gel - please use a different porosity testing method. [9] Fig. 4 and Table 1 please add the number of repetitions (n=x). [10] In the discussion, lacks a comparison of the selected oil with other data from the literature (other oils or bioactive substances used in gels). [11] Please add a section on Research limitations and future plans. [12] Conclusions should be precise!!; porosity has not been tested properly, you are unable to determine if it is favorable; sentence ''has antioxidant properties, anti-inflammatory activity, antimicrobial activity, and good healing activity'' must be changed to be substantively specific and contain accurate scientific results.
Author Response
Reviewer #1:
- The abstract should contain more precisely described the main effects of the research.
- Thanks for this suggestion. The reviewer considerations were considered. The abstract was evaluated and full corrected. The writing was corrected according to lines 14-33, page 1.
- Keywords should be also more precise.
- Thanks for this suggestion. The keywords were evaluated. The writing was corrected according to lines 34-35, page 1.
- In the Introduction, there is no presentation of other biomaterials dedicated to the treatment of wounds; the use of this oil in biomaterials; summaries of research to date on CH and CG gels.
- Thanks for this question. We appreciate the suggestions and improve the introduction. Lines 53-59; 66-70; 74-76; 83-87. Pages 2-3; 3; 4, respectively.
- You stated:[Polysacharides] They are biocompatibile, biodegradable, adhesive, non-toxic, antimicrobial, antioxidant and anti-inflammatory" - all of them?
- Thanks for this comment. Polysaccharides were obtained from plant, microbial, animal or algae sources have characteristics in common: biodegradability, low toxicity, biocompatibility, and renewable source. These characteristics make these materials with potential for replacing synthetic polymers, but not all natural polysaccharides have all properties, but some properties can be found in some of them. The text was changed by (lines 60-61, page 3): Some interesting properties can be found in some polysaccharides, especially biocompatible, biodegradable, adhesive, non-toxic, antimicrobial, antioxidant, and anti-inflammatory
- There is no description of the statistical analysis method.
- Thanks for this question. The statistical analysis method has been added according to lines 188-191, page 7.
- In the Materials and Methods, description of freeze-drying procedure is missing; justification for the selection of these bacteria / fungi supported by the literature.
- Thanks for this suggestion. The freeze-drying process was added, according to the lines 123-124, page 5. The microorganisms evaluated in this study are well known to be related to a variety of disseminated infections, including wounds. According to lines 273-276; 281-283, pages 12-13.
- In the Fig. 1, please also mark the describing in text groups (bonds, stretch etc.)
- We appreciate the suggestion. In fig. 1 and in the text, they are marked by arrows. Lines 198, 200, and 203. Pages 7-8.
- From these SEM images (Fig. 3), we are not able to determine adequately accurately the porosity of the gel - please use a different porosity testing method.
- Thanks for this question. In fig 2, unfortunately, we were not able to quantify the porosity. However, it is possible to observe the roughness and the presence of pores (qualitative analysis). They were marked with black arrows to improve visualization.
- 4 and Table 1 please add the number of repetitions (n=x).
- Thanks for this question. It was added in the materials and methods section that all experiments were carried out in triplicate. Lines 161 and 174. Pages 6-7
- In the discussion, lacks a comparison of the selected oil with other data from the literature (other oils or bioactive substances used in gels).
- Reviewer considerations have been considered, thanks. Literature data were added, according to the lines 278-280, 304-305, and 316-321. Page 12 and 14.
- Please add a section on Research limitations and future plans.
- We appreciate the suggestion, thanks, and add the limitations and perspectives. Lines 324 -326, Page 14.
- Conclusions should be precise!!; porosity has not been tested properly, you are unable to determine if it is favorable; sentence ''has antioxidant properties, anti-inflammatory activity, antimicrobial activity, and good healing activity'' must be changed to be substantively specific and contain accurate scientific results.
- We appreciate the suggestion, thanks, and the conclusion has been rewritten. Lines 338-344, page 15.

Reviewer 2 Report
Dear Authors,
I have read the mauscript and I think that it is a very well written manuscript with original preclinical data.
Even if I think that a traslational study supporting the effect of this compound in human could improve these data, this is not requested. Therefore I think that the manuscript is suitable for the publication
Author Response
Reviewer #2
I have read the mauscript and I think that it is a very well written manuscript with original preclinical data.
Even if I think that a traslational study supporting the effect of this compound in human could improve these data, this is not requested. Therefore I think that the manuscript is suitable for the publication
Thanks for the review.

Reviewer 3 Report
Dear Authors,
I have read with interest you manuscript and I think that it is very well written. I have only a minor question for you that is related to a previous manuscript that I have read on a similar topic, where the authors used quercetin (doi: 10.1111/iwj.13299). Please add it in references and in discussion
I think that the paper Is very well written and the data are very interesting. The mechanism of activity of each compound Is well descrivere and there are functional as well as cellular data supporting the effect of this preparation. As suggested to the Authors previously was reported an effect of a similar compound in both animals and humans. Clinical data could add data in safety but this Is not required for Natural products. However I think that this compound could be usefull in clinical practice even if human data could be of utility.Author Response
Reviewer #3
I have read with interest you manuscript and I think that it is very well written. I have only a minor question for you that is related to a previous manuscript that I have read on a similar topic, where the authors used quercetin (doi: 10.1111/iwj.13299). Please add it in references and in discussion
I think that the paper Is very well written and the data are very interesting. The mechanism of activity of each compound Is well descrivere and there are functional as well as cellular data supporting the effect of this preparation. As suggested to the Authors previously was reported an effect of a similar compound in both animals and humans. Clinical data could add data in safety but this Is not required for Natural products. However I think that this compound could be usefull in clinical practice even if human data could be of utility.
- Thanks for this considerations. We appreciate the reference suggestion and have been added. Line 321, page 14.
Round 2
Reviewer 1 Report
Manuscript ID: biomedicines-1660756
This manuscript, Potential wound healing effect of gel based on chicha gum, chitosan, and Mauritia flexuosa oil, is an interesting paper focusing on a novel gel based on two biopolymers incorporated with a bioactive oil. The paper is relatively well-thought, suitable for readers and shows new research results regarding wound healing. Moreover, the manuscript was revised and the authors responded fairly and accurately to all reviewers' comments. All indicated shortcomming were well corrected, hence I may suggest acceptance in present form.